# APOBEC3A deaminates transiently exposed single-strand DNA during LINE-1 retrotransposition

Sandra R Richardson[1]*, Iñigo Narvaiza[2], Randy A Planegger[1], Matthew D Weitzman[3], John V Moran[1,4]*

[1]Department of Human Genetics, University of Michigan Medical School, Ann Arbor, United States; [2]Laboratory of Genetics, The Salk Institute for Biological Studies, La Jolla, United States; [3]Department of Pathology and Laboratory Medicine, University of Pennsylvania Perelman School of Medicine and the Children's Hospital of Philadelphia, Philadelphia, United States; [4]Department of Internal Medicine, Howard Hughes Medical Institute, University of Michigan Medical School, Ann Arbor, United States

**Abstract** Long INterspersed Element-1 (LINE-1 or L1) retrotransposition poses a mutagenic threat to human genomes. Human cells have therefore evolved strategies to regulate L1 retrotransposition. The APOBEC3 (A3) gene family consists of seven enzymes that catalyze deamination of cytidine nucleotides to uridine nucleotides (C-to-U) in single-strand DNA substrates. Among these enzymes, APOBEC3A (A3A) is the most potent inhibitor of L1 retrotransposition in cultured cell assays. However, previous characterization of L1 retrotransposition events generated in the presence of A3A did not yield evidence of deamination. Thus, the molecular mechanism by which A3A inhibits L1 retrotransposition has remained enigmatic. Here, we have used in vitro and in vivo assays to demonstrate that A3A can inhibit L1 retrotransposition by deaminating transiently exposed single-strand DNA that arises during the process of L1 integration. These data provide a mechanistic explanation of how the A3A cytidine deaminase protein can inhibit L1 retrotransposition.

**\*For correspondence:** risandra@ umich.edu (SRR); moranj@umich. edu (JVM)

## Introduction

Long INterspersed Element-1 (LINE-1 or L1) derived sequences comprise roughly 17% of human genomic DNA (*Lander et al., 2001*). The vast majority of L1 sequences are 5′ truncated, internally rearranged, or are inactivated by mutations and are unable to mobilize (i.e., retrotranspose) to new genomic locations (*Grimaldi et al., 1984*; *Lander et al., 2001*). However, it is estimated that the average human genome contains 80–100 retrotransposition-competent (i.e., active) L1s (*Sassaman et al., 1997*; *Brouha et al., 2003*). Active human L1s are approximately 6 kb in length and contain a 5′ untranslated region (UTR), two open reading frames (ORF1 and ORF2), and a 3′ UTR that terminates in a poly (A) tail (*Scott et al., 1987*; *Dombroski et al., 1991*). ORF1 encodes an ~40 kDa protein (ORF1p) that binds L1 RNA as a trimer (*Holmes et al., 1992*; *Hohjoh and Singer, 1996*; *Martin et al., 2003*; *Khazina and Weichenrieder, 2009*). ORF1p also has nucleic acid chaperone activity (*Martin and Bushman, 2001*; *Khazina and Weichenrieder, 2009*). ORF2 encodes an ~150 kDa protein (ORF2p) (*Ergun et al., 2004*; *Doucet et al., 2010*; *Taylor et al., 2013b*) with DNA endonuclease (*Feng et al., 1996*) and reverse transcriptase (*Mathias et al., 1991*) activities. The development of a cultured cell assay definitively demonstrated that both ORF1p and ORF2p are required for efficient L1 retrotransposition (*Moran et al., 1996*). Subsequent biochemical studies revealed that

**eLife digest** Transposable elements are often referred to as 'jumping genes' because they can move between different locations within a genome. These sequences of DNA are found in many organisms and can make up a significant proportion of the genetic material: almost 50% of the DNA in the case of the human genome.

Transposable elements are grouped by how they move to new locations in a genome. Some move by a cut-and-paste mechanism—whereby the transposable element DNA is removed from one location and inserted back at a new genomic location. Others, termed retrotransposons, move by a copy-and-paste mechanism: the DNA sequence is transcribed into an RNA intermediate, and then copied back into DNA before being inserted into a new location. Retrotransposons can accumulate to great numbers in genomes: and one retrotransposon, called LINE-1, is present at an estimated 500,000 copies in the human genome.

Although most copies of LINE-1 are inactive, the average human genome contains about 80–100 that are predicted to be able to 'jump' to new locations. Given that these retrotransposons could insert into, and disrupt, vital genes, it follows that our cells would have evolved ways to limit their movement. An enzyme named APOBEC3A is known to limit the movement of LINE-1 retrotransposons in cells. APOBEC3A can alter the letters, or bases, that make up the genetic code. This enzyme acts on single-strand DNA to change 'C' bases to 'U' bases, which could explain how APOBEC3A combats LINE-1. However, no evidence for such mutation of LINE-1 sequences by APOBEC3A had been found to date.

Now, Richardson et al. recreate the copying of LINE-1 RNA back into DNA in a test tube—and reveal that APOBEC3A can mutate single-strand LINE-1 DNA. Critically, as long as the RNA intermediate and DNA copy remain together, the LINE-1 DNA is protected. However, when LINE-1 inserts into a new location the temporarily exposed single strand of LINE-1 DNA becomes susceptible to mutation by APOBEC3A. Human cells can detect and destroy 'U' bases in DNA—and only by inhibiting this ability were Richardson et al. able to observe APOBEC3A mutations in new LINE-1 copies within the genomes of living cells.

Richardson et al. speculate that the activity of APOBEC3 enzymes must strike a balance between limiting the spread of retrotransposons and minimizing the mutation of the cell's own DNA. Future work could address important questions, such as: do APOBEC enzymes affect the 'jumping' of LINE-1 retrotransposons in human reproductive cells and the early embryo, where new LINE-1 insertions could be passed on to subsequent generations? Also, does a loss of APOBEC3 activity lead to new LINE-1 insertions in cancerous cells? And does this effect how tumors form and/or progress? Since APOBEC3 enzymes can cause mutations in cancers, they have been proposed as new targets for anti-cancer drugs—therefore, it is crucial to uncover any harmful effects of inhibiting APOBEC3 enzymes that might limit the effectiveness of such treatments.

L1 retrotransposition occurs by a mechanism termed target-site primed reverse transcription (TPRT) (*Luan et al., 1993*; *Feng et al., 1996*; *Cost et al., 2002*; *Kulpa and Moran, 2006*).

Ongoing L1 retrotransposition events contribute to intra- and inter-individual genetic variation and, on occasion, can disrupt gene function, leading to sporadic genetic diseases (*Beck et al., 2011*; *Hancks and Kazazian, 2012*). Given the mutagenic potential of active L1s, it is not surprising that host cells have evolved multiple mechanisms to restrict L1 retrotransposition (*Levin and Moran, 2011*). Previous studies revealed that over-expression of the wild-type (WT) APOBEC3A cytidine deaminase protein (A3A), but not deaminase-defective A3A mutants, could potently inhibit L1 retrotransposition in cultured human cells (*Chen et al., 2006*; *Muckenfuss et al., 2006*; *Bogerd et al., 2006b*; *Kinomoto et al., 2007*; *Niewiadomska et al., 2007*). However, deamination events were not detected in L1 retrotransposition events generated in the presence of WT A3A (*Chen et al., 2006*; *Muckenfuss et al., 2006*; *Bogerd et al., 2006b*; *Kinomoto et al., 2007*). Thus, the molecular mechanism by which WT A3A inhibits L1 retrotransposition has remained a mystery.

Here, we employ an in vitro assay (L1 Element Amplification Protocol or LEAP) that recapitulates the first strand cDNA synthesis step of the L1 integration to elucidate a mechanism for A3A-mediated inhibition of L1 retrotransposition. In LEAP reactions that include RNase H, an enzyme that specifically

degrades the RNA strand of an RNA/DNA heteroduplex, we report that single-strand L1 cDNAs are rendered susceptible to A3A-mediated deamination. These data suggest that newly synthesized L1 cDNAs remain annealed to their mRNA templates after reverse transcription, protecting the L1 cDNA from deamination by A3A. We further demonstrate that engineered L1 retrotransposition events derived from cultured human cells that co-express A3A and the uracil DNA glycosylase inhibitor protein (UGI) contain strand-specific, A3A-mediated C-to-U deamination events. Together, these data indicate that A3A can inhibit L1 retrotransposition by deaminating transiently exposed single-strand DNA that arises during the process of L1 integration. Moreover, our results provide direct evidence that A3A can deaminate single-strand genomic DNA in human cells.

## Results

We used a cultured cell assay (*Moran et al., 1996*; *Wei et al., 2000*) to determine the extent to which A3A inhibits retrotransposition of LINE elements that differ significantly in their nucleotide sequence and structure (*Figure 1A*). The retrotransposition indicator cassette mneoI (*Freeman et al., 1994*; *Moran et al., 1996*) was used to tag the 3′ untranslated regions (UTRs) of a human L1 (pJM101/L1.3) (*Sassaman et al., 1997*), a natural mouse L1 element (TGF21) (*Goodier et al., 2001*), a synthetic mouse L1 element (L1SM) (*Han and Boeke, 2004*), and a LINE-2 element from zebrafish (ZfL2-2) (*Sugano et al., 2006*; *Figure 1A*). The mneoI cassette consists of an antisense neomycin phosphotransferase gene (NEO) disrupted by an intron, which is in the same transcriptional orientation as the LINE element (*Figure 1A*). This arrangement ensures that G418-resistant foci arise only if the tagged LINE element undergoes successful retrotransposition (*Freeman et al., 1994*; *Moran et al., 1996*). HeLa cells do not express endogenous A3A (*Wissing et al., 2011*), and therefore provide a system to measure directly the effects of ectopically expressed A3A on L1 retrotransposition.

We co-transfected HeLa cells with retroelement expression constructs and plasmids encoding WT A3A or a deaminase-deficient mutant, A3A-C106S (*Chen et al., 2006*). As a control we included β-arrestin, which does not significantly affect L1 retrotransposition (*Bogerd et al., 2006b*). A3A expression inhibited human L1.3 retrotransposition to ~28% of control levels (*Figure 1B*), in agreement with previous reports (*Chen et al., 2006*; *Muckenfuss et al., 2006*; *Bogerd et al., 2006b*). In comparison to control levels, A3A also inhibited retrotransposition of TGF21 (to ~20%), ZfL2-2 (to ~17%), and L1SM (to ~61%) (*Figure 1B*). The milder inhibition of A3A on L1SM retrotransposition may be due to the GC-rich nature of the L1SM transcript, the elevated steady state levels of L1SM mRNA, and/or an increased number of L1SM retrotransposition events per cell (*Han and Boeke, 2004*). Additional controls revealed that the reduction of G418-resistant HeLa cell foci observed in the presence of A3A reflects specific inhibition of L1 retrotransposition, rather than non-specific cytotoxicity (*Figure 1—figure supplement 1A–C*; see Methods and Materials). A deaminase-deficient mutant, A3A-C106S, did not significantly restrict retroelement mobility (*Figure 1B*), in agreement with previous studies (*Chen et al., 2006*; *Muckenfuss et al., 2006*; *Bogerd et al., 2006b*). Moreover, A3A inhibited L1.3 retrotransposition when it was expressed from its native 5′UTR on a non-episomal vector (*Figure 1—figure supplement 1D*: pKS101/L1.3/sv+). Together, these experiments show that A3A inhibits retrotransposition of diverse LINE elements in cultured cells.

Since LINE element retrotransposition proceeds by target-site primed reverse transcription (TPRT), which requires element-encoded endonuclease (EN) and reverse transcriptase (RT) activities (*Luan et al., 1993*; *Feng et al., 1996*), we investigated whether A3A specifically interferes with L1 EN or L1 RT activity. In addition to canonical TPRT, human L1s can mobilize by an alternative, endonuclease-independent retrotransposition mechanism (ENi) in cells that lack p53 activity and are defective for components of the non-homologous end joining (NHEJ) DNA repair machinery (*Morrish et al., 2002*, *2007*; *Coufal et al., 2011*). We reasoned that if A3A specifically inhibits L1 EN activity, ENi L1 retrotransposition events should escape A3A inhibition. To test this hypothesis, we co-transfected a human L1 (either WT pAD2TE1 or EN-deficient pAD136 [*Doucet et al., 2010*]) and a WT A3A expression construct into Chinese Hamster Ovary (CHO) cell lines that are NHEJ-proficient (4364a) or NHEJ-deficient (XR-1) (*Morrish et al., 2002*, *2007*). A3A expression inhibited WT L1 retrotransposition in both cell lines, and also inhibited ENi retrotransposition in XR-1 cells (*Figure 1C*). These data demonstrate A3A can inhibit L1 retrotransposition in a hamster cell line and that the A3A-mediated reduction of L1 retrotransposition is not due to inhibition of L1 EN activity.

To test whether A3A inhibits L1 RT activity, we performed in vitro LEAP assays (*Kulpa and Moran, 2006*) with ribonucleoprotein particle (RNP) preparations derived from cells transfected with an

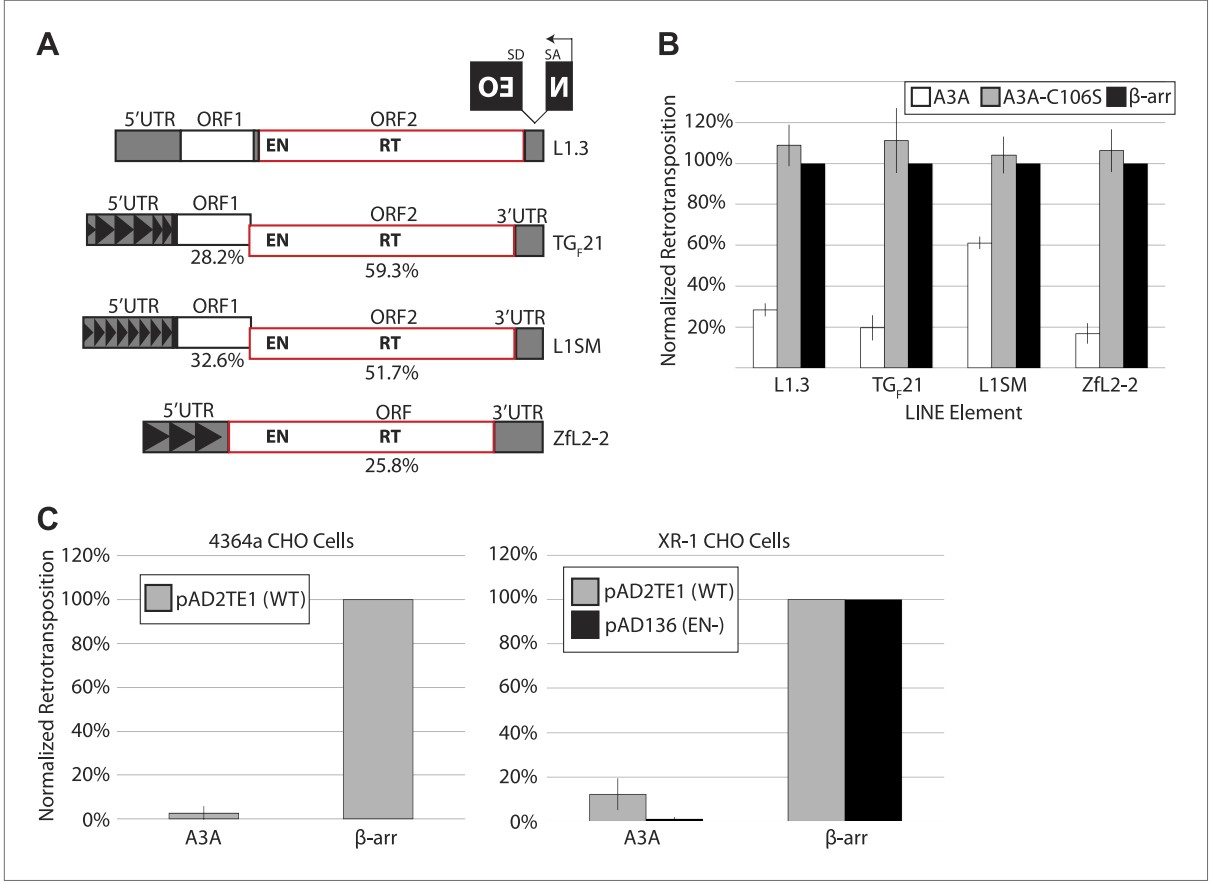

**Figure 1**. A3A suppresses retrotransposition of distinct LINEs, but does not specifically inhibit L1 endonuclease activity. (**A**) *LINE schematics*: L1.3, TG$_F$21, and L1SM encode two ORFs (ORF1 and ORF2, outlined in black and red, respectively). ZfL2-2 contains one ORF (outlined in red). Indicated are the endonuclease (EN) and reverse transcriptase (RT) domains in the respective ORFs. All elements are tagged in their 3'UTRs with the *mneoI* (NEO) indicator cassette. The percent nucleotide identity of TG$_F$21, L1SM, and ZfL2-2 to human L1.3 is indicated below each schematic. Black arrowheads indicate repeated monomeric sequences in the 5' UTRs of TG$_F$21, L1SM, and ZfL2-2. (**B**) *A3A inhibits retrotransposition of distinct LINEs*: shown is the effect of wild-type A3A (white bars), deaminase-deficient A3A_C106S (gray bars), and β-arrestin (β-arr) control (black bars) on LINE retrotransposition. The x-axis indicates the LINE element. The y-axis indicates percent retrotransposition; in each case the β-arr control is set to 100%. Data were normalized using a circular *NEO* expression cassette as detailed in *Figure 1—figure supplement 1A*. Data are expressed as the mean percent retrotransposition derived from three independent experiments consisting of two technical replicates each, with error bars representing the standard deviation among all six technical replicates. (**C**) *Retrotransposition assays in 4364a (wild-type CHO cells (left)) and XR-1 (XRCC4-deficient CHO cells (right))*: the x-axis indicates experiments conducted with the A3A or β-arr expression vector. The y-axis indicates percent retrotransposition; for each reaction the β-arr control is set to 100%. Gray bars indicate retrotransposition of wild-type L1 (pAD2TE1) and black bars indicate the L1 EN mutant (pAD136). Data were normalized using the circular *NEO* control (*Figure 1—figure supplement 1A*). Data are shown as the mean percent retrotransposition derived from three independent experiments consisting of two or three technical replicates each, with error bars representing the standard deviation among all eight technical replicates.

The following figure supplements are available for figure 1:

**Figure supplement 1**. Additional Control Experiments.

engineered L1 construct (*Figure 2A*). We purified recombinant WT A3A (rA3A) (*Figure 2—figure supplement 1A,B*) and deaminase-deficient (rA3A_C106S) (data not shown) proteins from *Escherichia coli*. As expected rA3A had cytidine deaminase activity on single-strand (*Figure 2B*), but not on double-stranded DNA (*Figure 2—figure supplement 1C*), whereas rA3A_C106S lacked appreciable deaminase activity (*Figure 2B*). Incorporating increasing amounts of rA3A or rA3A_C106S into LEAP or Moloney murine leukemia virus (MMLV) RT reactions did not significantly affect the synthesis of L1 or MMLV-derived cDNA products (*Figure 2C*, *Figure 2—figure supplement 1D*). Thus, A3A does not specifically block L1 RT activity.

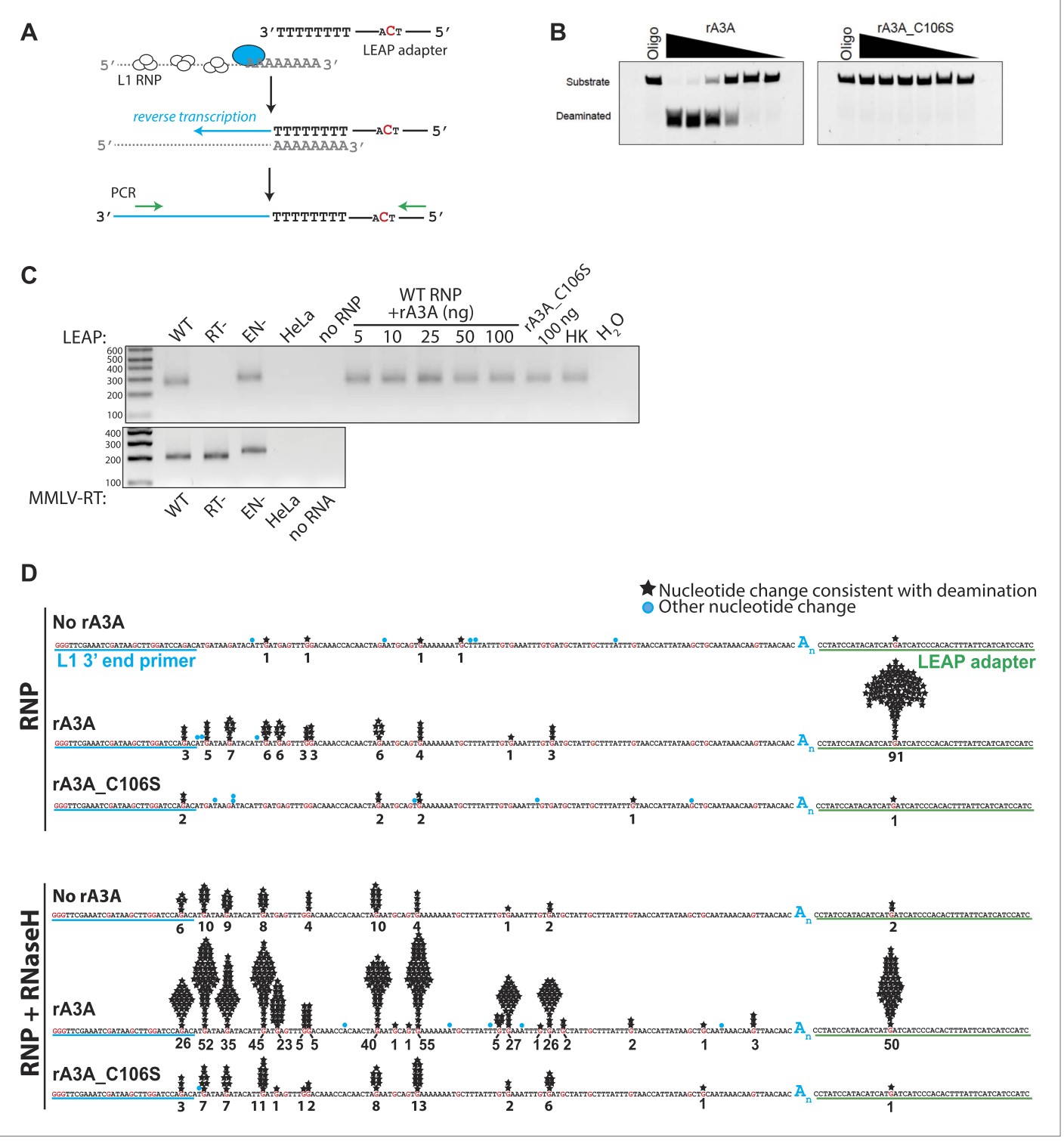

**Figure 2**. Recombinant A3 (rA3A) deaminates L1 cDNAs in vitro. (**A**) *LEAP assay rationale*: L1 RNP preparations consisting of the L1 RNA (gray), L1 ORF1p (white ovals), and L1 ORF2p (blue oval) are incubated with a 3′ RACE primer consisting of a unique adapter sequence that contains a single cytidine (red) followed by an oligo dT sequence (black lettering). After reverse transcription (blue arrow), the resultant L1 cDNAs (blue line) are PCR amplified using primers specific to the engineered L1 and the unique adapter sequence (green arrows). (**B**) *Recombinant A3A has deaminase activity in vitro*: twofold serial dilutions (500 ng–15.62 ng) of WT rA3A (left panel) or deaminase-deficient rA3A_C106S (right panel) were incubated with a fluorescein isothiocyanate (FITC) labeled single-strand DNA oligonucleotide containing a single cytidine residue. The products were treated with recombinant

*Figure 2. Continued on next page*

*Figure 2. Continued*

uracil DNA glycosylase (UDG) and NaOH and then were resolved by gel electrophoresis. A control reaction was included without recombinant protein (marked Oligo). (**C**) *Recombinant A3A does not inhibit L1 RT activity*: control LEAP reactions with RNP preparations from HeLa cells transfected with WT (pDK101), RT- (pDK135), or EN- (pJJH230A/L1.3) human L1s (upper gel). HeLa indicates untransfected HeLa cells; no RNP indicates control reactions lacking RNPs. Increasing amounts of rA3A (ng) did not significantly affect LEAP activity. Samples containing a deaminase-deficient rA3A_C106S, a heat killed rA3A (HK), or without LEAP products ($H_2O$) served as controls. MMLV RT reactions (lower gel) confirm the integrity of purified RNA isolated from RNP preps used in the LEAP assay. Notably, the increased size of the EN- RNP RT products is due to a higher molecular weight product generated from pJJH230A/L1.3, which contains an *mblastl* indicator cassette instead of an *mneol* indicator cassette. Size standards (bp) are indicated at the left of the gel. (**D**) *Sequence characterization of LEAP Products*: shown is the (+) strand sequence of the LEAP product. Guanosine nucleotides are indicated in red. Black stars and numbers indicate the frequency of G-to-A mutations (corresponding to C-to-U mutations in the minus (−) strand L1 cDNA) that occurred on (+) strand L1 cDNA. Blue circles indicate other nucleotide changes. The blue $A_n$ indicates the LEAP product poly (**A**) tail. Blue underlining indicates the L1 3′ end PCR primer. Green underlining indicates the LEAP adapter (5np1) sequence. *Top panel*: LEAP products generated under the following conditions: no rA3A protein, 100 ng of wild-type rA3A, 100 ng of deaminase-deficient rA3A_C106S. *Bottom panel*: LEAP products generated under the following conditions in the presence of RNase H: no rA3A protein, 100 ng of wild-type rA3A, 100 ng of deaminase-deficient rA3A_C106S. One hundred products were characterized for each condition.

The following figure supplements are available for figure 2:

**Figure supplement 1**. Control Experiments with Recombinant A3A.

**Figure supplement 2**. A3A Deamination Events in LEAP products.

**Figure supplement 3**. Distribution of deamination events per LEAP product.

**Figure supplement 4**. Summary of LEAP products generated in the presence of rA3A and RNase H.

To determine whether A3A can deaminate L1 cDNAs in vitro, we sequenced 100 L1 cDNA products from LEAP reactions conducted in the absence of A3A (no rA3A) or in the presence of rA3A or deaminase-deficient rA3A_C106S. We did not detect significant numbers of C-to-U deamination events in L1 cDNAs derived from LEAP reactions in the absence of rA3A or the presence of rA3A_C106S (*Figure 2D*). In contrast, LEAP products generated in the presence of rA3A contained abundant C-to-U editing that was most frequent at the single 5′-TCA-3′ consensus A3A deamination site (*Chen et al., 2006*) present in the single-strand oligonucleotide adapter used to prime L1 RT reactions (*Figure 2D*, *Figure 2—figure supplement 2*, top chart in the middle panel, 91 of 100 (91%) total 5′-TCA-3′ sites within the 100 single-strand oligonucleotide adapters showed evidence of deamination). However, far less deamination was detected within L1 cDNA LEAP products at the six internal 5′-TCA-3′ sites (*Figure 2D*, *Figure 2—figure supplement 2*, top chart in the middle panel, only 25 of 600 (4.2%) 5′-TCA-3′ sites within the 100 L1 cDNA products showed evidence of deamination).

The non-uniform pattern of deamination events observed above suggested that the L1 cDNA may remain annealed to L1 mRNA, generating L1 mRNA/cDNA heteroduplexes after reverse transcription, thereby leaving only the single-strand oligonucleotide LEAP adapter susceptible to deamination. Thus, we tested whether inclusion of RNase H into LEAP reactions renders L1 cDNA susceptible to deamination. In the presence of rA3A and RNase H, 228 of 600 (38%) 5′-TCA-3′ substrates within the L1 cDNAs showed evidence of deamination (*Figure 2D*, *Figure 2—figure supplement 2*, bottom chart in the middle panel). Importantly, sequencing revealed differing poly (A) tail lengths and unique deamination patterns in each of the 100 LEAP products, indicating that they were derived from independent L1 cDNAs (*Figure 2—figure supplement 3*, panel e; *Figure 2—figure supplement 4*). In contrast, in the presence of rA3A_C106S and RNase H, only 6.7% of 5′-TCA-3′ substrates within 100 L1 cDNAs demonstrated evidence of deamination (*Figure 2D*, *Figure 2—figure supplement 2*, bottom chart in the right panel). Similarly, in reactions conducted with RNase H alone, 4.2% of 5′-TCA-3′ substrates within 100 L1 cDNAs contained signatures of deamination (*Figure 2D*, *Figure 2—figure supplement 2*, bottom chart in the left panel). The slightly increased number of deamination events observed with RNase H alone may reflect endogenous cytidine deaminase activity present in HeLa RNP preparations. APOBEC3B (A3B) might be responsible for these events as it is expressed in HeLa cells and can suppress L1 retrotransposition (*Chen et al., 2006*; *Bogerd et al., 2006b*; *Wissing et al.,*

2011). These results suggest that the L1 cDNA, when annealed to the L1 mRNA in an mRNA/cDNA heteroduplex, is protected from mutation by cellular deaminases.

Our results demonstrated that rA3A could deaminate single-strand L1 cDNAs in vitro. We therefore hypothesized that A3A may deaminate transiently exposed single-strand DNA intermediates that arise during L1 TPRT in cultured cells. Uridine residues are efficiently removed from DNA through combined actions of uracil DNA glycosylase (UNG) and apurinic/apyrimidinic endonuclease (APE) (*Krokan et al., 2002*). In principle, these activities could repair and/or degrade deaminated single-strand DNAs that arise during TPRT. Degradation of deaminated retrotransposition intermediates would reduce L1 retrotransposition levels in cultured cell assays and could, in principle, eliminate evidence of A3A-mediated deamination events in the resultant retrotransposed L1 sequences. Such a scenario could explain why previous studies failed to detect A3A-mediated deamination events in engineered L1 retrotransposition events (*Chen et al., 2006*; *Muckenfuss et al., 2006*; *Bogerd et al., 2006b*; *Kinomoto et al., 2007*).

To test whether UNG contributed to the inhibition of L1 retrotransposition downstream of A3A activity, we conducted L1 retrotransposition assays in the presence of codon optimized bacteriophage-derived uracil DNA glycosylase inhibitor (UGI) that selectively inhibits UNG activity (*Kaiser and Emerman, 2006*; *Figure 3A*). Remarkably, UGI expression from a transfected plasmid (pLGCX_UGI) alleviated A3A-mediated L1 inhibition by ~twofold (*Figure 3B*: from ~25% to ~57% of control levels). In contrast, UGI expression had little effect on L1 retrotransposition activity in assays conducted in the presence of A3A deaminase mutant (A3A-C106S) or β-arrestin control (*Figure 3B*). Similarly, UGI expression did not affect the ability of deaminase-deficient A3B mutants (*Bogerd et al., 2006b*) to inhibit L1 retrotransposition (*Figure 3—figure supplement 1A*: see A3B_N-term and A3B_CS data). Thus, UGI expression specifically alleviates a deaminase-dependent mechanism of L1 retrotransposition inhibition.

We hypothesized that UGI expression should allow the detection of A3A-mediated C-to-U deamination events in retrotransposed L1s. Thus, we co-transfected HeLa cells with A3A, UGI, and pJM140/L1.3/Δ2/k7, an engineered L1 vector that contains both an mneoI retrotransposition cassette and a ColE1 bacterial origin of replication (*Gilbert et al., 2002*, *2005*) (*Figure 3A*). This modified cassette facilitates recovery of engineered L1 retrotransposition events and their associated flanking genomic DNA sequences as autonomously replicating, kanamycin-resistant plasmids in bacteria (*Gilbert et al., 2002*, *2005*). Characterization of 12 independent retrotransposition events revealed four G-to-A mutations on the L1 coding strand, which correspond to C-to-U deamination events on the (−) strand L1 cDNA (*Figure 3C*). Notably, the 5' target-site duplication (TSD) of one insertion contained a C-to-T change within a 5'-TCA-3' consensus A3A deamination site in flanking genomic DNA. This putative A3A-mediated deamination event led to an inexact TSD flanking a newly retrotransposed L1 (*Figure 3D*).

To gain further evidence for A3A-mediated deamination of L1 retrotransposition events, we employed a stable UGI-expressing U2OS cell line (*Landry et al., 2011*) (*Figure 3A*: right side). This strategy ensured that UNG activity is inhibited before co-transfection of L1 and A3A expression plasmids (*Landry et al., 2011*). In control U2OS cells, transfection of a low amount of A3A plasmid (25 ng) inhibited retrotransposition to ~28.0% of control levels without substantial cytotoxicity (*Figure 3E*, *Figure 3—figure supplement 1B*). By comparison, in U2OS_UGI-expressing cells, A3A only slightly inhibited L1 retrotransposition (*Figure 3E*: to ~83% of control levels; *Figure 3—figure supplement 1C*). Thus, consistent with experiments performed in HeLa cells, stable UGI expression relieves A3A-mediated inhibition of L1 retrotransposition in U2OS cells.

We next analyzed sequences of engineered L1 retrotransposition events. We used pJM140/L1.3/Δ2/k7 to generate a panel of clonal G418-resistant U2OS_UGI cell lines, isolated the resultant L1 retrotransposition events, and examined them for evidence of A3A-mediated deamination. Characterization of 24 control L1 retrotransposition events did not reveal any G-to-A mutations (*Figure 3F*, top panel, *Figure 3—figure supplement 2*). Similarly, characterization of 30 L1 retrotransposition events generated in the presence of A3A-C106S revealed only five G-to-A mutations (*Figure 3F*, middle panel, *Figure 3—figure supplement 3*). In contrast, characterization of 33 L1 retrotransposition events generated in the presence of A3A revealed 39 G-to-A mutations (*Figure 3F*, bottom panel, *Figure 3—figure supplement 4*). The A3A-induced mutations occurred in a strand-specific manner (i.e., we only detected one C-to-T mutation on the L1 coding strand), and far outnumbered any other nucleotide changes in retrotransposed products. Notably, one mutation (pJM140/L1.3/Δ2/k7: G-5390-A) introduced a premature termination codon into L1 ORF2, which would create

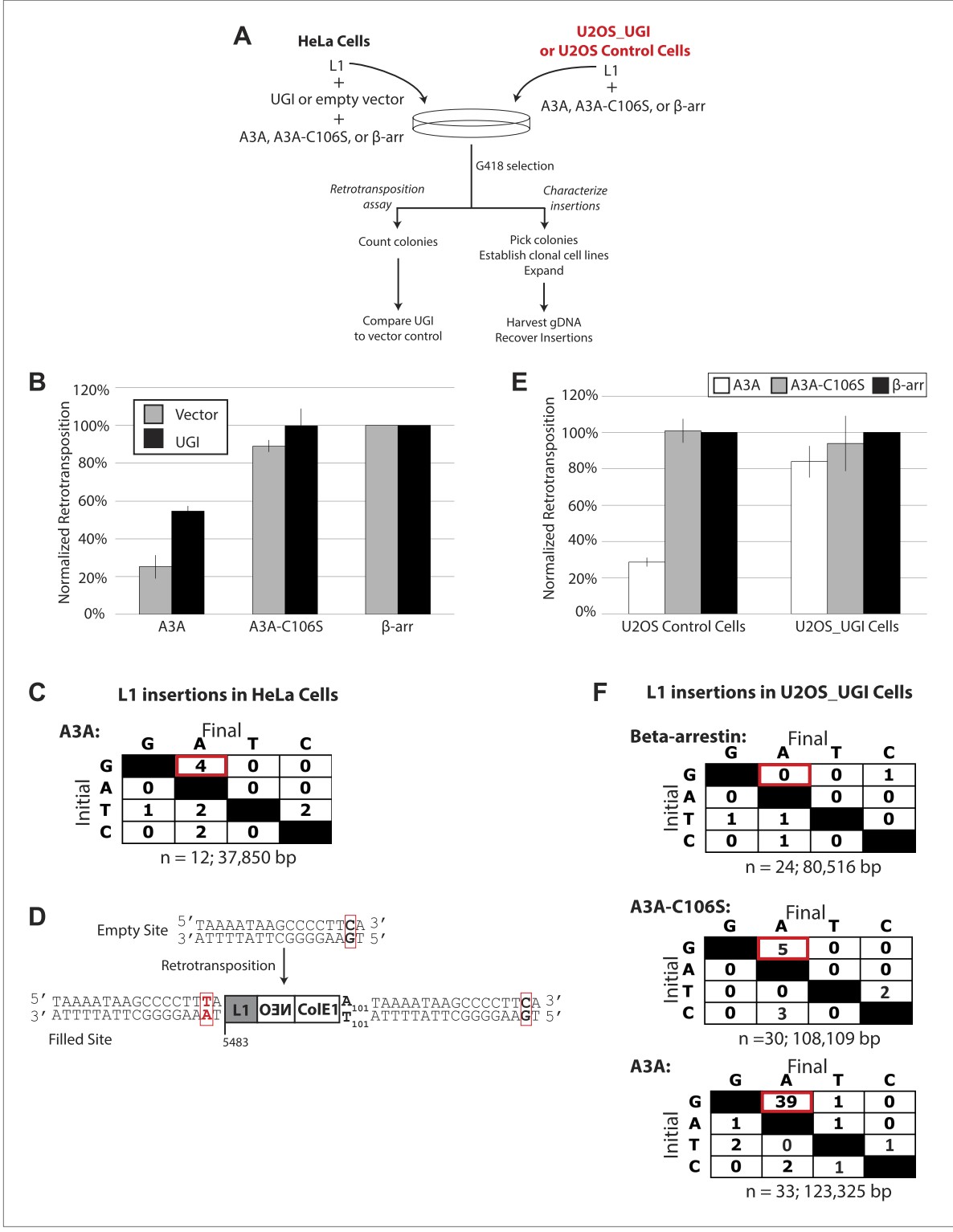

**Figure 3**. UGI expression alleviates A3A-mediated inhibition of L1 retrotransposition and allows detection of deaminated L1 retrotransposition events. (**A**) *Experimental strategy*: cells (HeLa, upper left, or U2OS, upper right) were co-transfected with the indicated expression plasmids and assayed for L1 retrotransposition (bottom left branch) or insertion analysis (bottom right branch). (**B**) *UGI expression alleviates A3A-mediated retrotransposition inhibition*: the x-axis indicates the co-expression vector. The y-axis depicts the efficiency of L1 retrotransposition. Shown are results of experiments in the presence (black bars) or absence (gray bars) of UGI. The results were normalized as in *Figure 1—figure supplement 1A*. Data are expressed as the mean percent retrotransposition derived from three independent experiments consisting of two technical replicates each, with error bars representing
*Figure 3. Continued on next page*

*Figure 3. Continued*

the standard deviation among all six technical replicates. (**C**) *DNA sequencing results*: the left column indicates the engineered L1 sequence. The top row indicates sequence changes observed in recovered L1 insertions; n indicates the number of characterized retrotransposition events. The total retrotransposed sequence observed (in bp) includes the L1 sequence and the *mneol/ColE1 cassette*. (**D**) *An L1 insertion harboring mismatched TSDs*: The pre- (empty) and post- (filled) L1 integration sites are shown. The truncation point (bp 5483) and structure of the engineered retrotransposed L1 are indicated. Deamination of single-strand genomic DNA in the pre-integration site (red rectangle) leads to inexact TSDs in the post-integration site (red lettering/rectangles). (**E**) *Stable UGI expression alleviates A3A-mediated L1 inhibition*: The x-axis indicates the U2OS cell line. The y-axis depicts the efficiency of L1 retrotransposition. Shown are the effects of wild-type A3A (white bars), A3A-C106S (gray bars), and β-arrestin (β-arr, black bars) control on L1 retrotransposition. Data were normalized to controls conducted with a circular *NEO* expression vector (***Figure 3—figure supplement 1 B–C***). Data are expressed as the mean percent retrotransposition derived from three independent experiments consisting of two technical replicates each, with error bars representing the standard deviation among all six technical replicates. (**F**) *DNA sequencing results*: The left column indicates the engineered L1 sequence. The top row indicates sequence changes observed in recovered L1 insertions; n indicates the numbers of characterized retrotransposition events. The total retrotransposed sequence observed (in bp) includes the L1 sequence and the *mneol/ColE1 cassette*.

The following figure supplements are available for figure 3:

**Figure supplement 1**. Additional Control Experiments.

**Figure supplement 2**. Summary of L1 retrotransposition events generated in U2OS_UGI cells in the presence of β-arrestin.

**Figure supplement 3**. Summary of L1 retrotransposition events generated in U2OS_UGI cells in the presence of A3A_C106S.

**Figure supplement 4**. Summary of L1 retrotransposition events generated in U2OS_UGI cells in the presence of A3A.

**Figure supplement 5**. The L1.3 G-5390-A mutation decreases L1 retrotransposition efficiency.

a 145 amino acid truncation in ORF2p. This mutation reduced L1 retrotransposition efficiency to ~10% of control levels (***Figure 3—figure supplement 5***). Thus, coupled with identification of a deamination-induced mutation in genomic DNA flanking a newly retrotransposed L1 (***Figure 3D***), these data suggest that the detected A3A-mediated deamination events occur on single-strand DNA during TPRT rather than on L1 mRNA or single-strand episomal plasmid DNA prior to L1 retrotransposition.

## Discussion

We propose a straightforward model for how A3A inhibits L1 retrotransposition (***Figure 4***). In the presence of cellular RNase H, the (−) strand L1 cDNA becomes vulnerable to A3A deamination (***Figure 4A***). Normally, these events would be repaired by combined actions of UNG and APE, which we speculate could lead to the integration of a 5′ truncated L1 cDNA that lacks evidence of deamination. However, in the presence of ectopically expressed UGI, A3A-mediated deamination events become unveiled in the retrotransposed L1s (***Figure 3D,F, 4A***). We further propose that A3A can act on transiently exposed single-strand genomic DNA flanking L1 integration sites (***Figure 4B,C***). The deamination of 5′-flanking top-strand genomic DNA could account for generation of inexact target-site duplications (***Figure 3D***). Interestingly, the structure of the proposed L1 integration intermediate resembles R-loop intermediates, formed during immunoglobulin class switching, that are targeted by an A3A related enzyme, activation-induced cytidine deaminase (AID) (***Goodman et al., 2007***). The deamination of 3′ flanking bottom-strand genomic DNA, in principle, could also lead to the generation of inexact TSDs (***Figure 4C***). While we did not observe these events in cultured cells, we did observe deamination events in the single-strand DNA oligonucleotide primer used to simulate 3′ flanking genomic DNA in LEAP assays (***Figure 2D***). It is intriguing to speculate that, in the absence of UGI, A3A-mediated editing within 3′ flanking bottom-strand genomic DNA could ultimately lead to loss of the L1 cDNA. Indeed, the ERCC1/XPF endonuclease has been implicated in limiting L1 retrotransposition by cleaving the transiently exposed single-strand 3′ flanking bottom-strand genomic DNA during TPRT (***Gasior et al., 2008***). The DNA lesion resulting from cleavage of this region could then be repaired using the top-strand genomic DNA as a template, leaving no detectable evidence of thwarted L1 retrotransposition events.

The over-expression of A3A does not abolish L1 retrotransposition completely, but reduces it to ~30% of control levels. Moreover, some L1 retrotransposition events generated in the presence of A3A

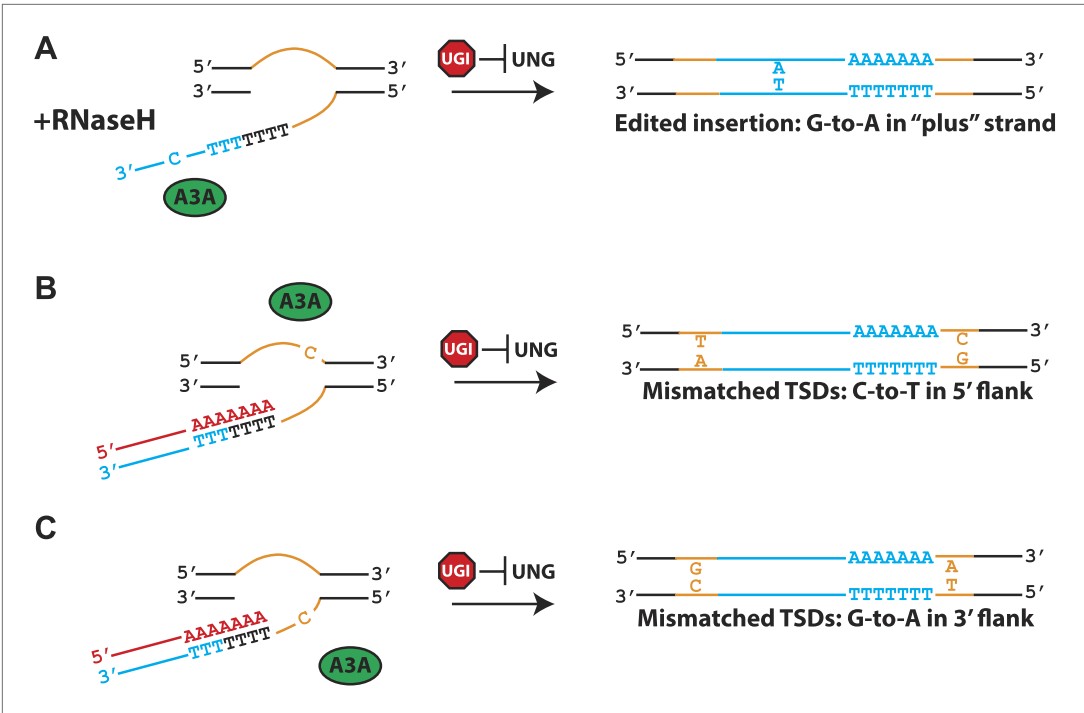

**Figure 4**. A model for A3A-mediated inhibition of L1 retrotransposition. The left side of the Figure shows L1 integration reactions that occur via TPRT. The right side of the Figure shows the predicted structures of the resultant L1 retrotransposition events. Shown are transiently exposed single-strand genomic DNA regions that ultimately give rise to target site duplications (TSDs: orange lines), the L1 RNA (red line), the L1 cDNA (blue line), and the A3A protein (green oval). UGI expression (red stop sign) can inhibit UNG activity. (**A**) *Deamination of the L1 cDNA*: in the presence of cellular RNase H, A3A-mediated deamination of the L1 (−) strand cDNA (blue lettering) in the presence of UGI leads to C-to-T mutations on the L1 non-coding strand and G-to-A mutations on the L1 (+) coding strand. (**B**) *Deamination of the 5′ flanking top-strand genomic DNA*: in the presence of UGI, deamination of the transiently exposed single-strand 5′ flanking genomic DNA (top orange line) during TPRT results in C-to-T changes in the 5′ TSD relative to the 3′ TSD. (**C**) *Deamination of the 3′ flanking bottom-strand genomic DNA*: in the presence of UGI, deamination of transiently exposed single-strand 3′ flanking genomic DNA during TPRT, in principle, is predicted to result in a G-to-A change in the 3′ TSD relative to the 5′ TSD.

and UGI lacked evidence of deamination. Why and how some L1 retrotransposition events evade A3A-mediated deamination requires further study. However, it is possible that ectopic A3A expression is too low to efficiently target L1 TPRT intermediates in some cells or that L1 second-strand cDNA synthesis occasionally is completed before A3A can gain access to TPRT intermediates to deaminate single-strand L1 cDNAs.

In sum, we conclude that deamination of transiently exposed single-strand DNA arising during L1 TPRT is responsible, at least in part, for A3A-mediated inhibition of L1 retrotransposition. It is noteworthy that the mobility of an evolutionarily related bacterial group II intron requires RNase H1 activity (*Smith et al., 2005*). Similarly, the related Bombyx mori R2 retrotransposon RT can displace RNA from RNA/DNA hybrids during second-strand cDNA synthesis in vitro (*Kurzynska-Kokorniak et al., 2007*). We hypothesize that annealed L1 mRNA protects the first-strand L1 cDNA, and that cellular RNase H renders these L1 cDNAs vulnerable to A3A deamination during TPRT.

Mechanistic insights gained from studying deaminase-dependent A3A-mediated L1 inhibition could be applicable to retroelement inhibition by other A3 proteins. A3A expression is restricted to peripheral blood lymphocytes (*Chen et al., 2006*; *Refsland et al., 2010*; *Stenglein et al., 2010*; *Thielen et al., 2010*; *Berger et al., 2011*; *Koning et al., 2011*), suggesting that it does not play a physiological role in restricting heritable L1 retrotransposition events. By comparison, A3B, which like A3A localizes to the nucleus, is expressed in human embryonic stem cells (hESCs) (*Bogerd et al., 2006b*) and can restrict engineered L1 retrotransposition in hESCs (*Wissing et al., 2011*). Since

heritable L1-mediated retrotransposition events can occur in the early human embryo (*de Boer et al., 2014*; *van den Hurk et al., 2007*), A3B may restrict heritable L1 retrotransposition events. Consistent with this notion, recent reports suggest that A3B is expressed at higher levels in human induced pluripotent stem cells (iPSCs) when compared to non-human primate iPSCs, and that A3B expression may, in part, account for the lower number of recent L1 insertions in the human genome compared to the chimpanzee genome (*Marchetto et al., 2013*). Intriguingly, an A3B deletion allele also is present at a high allele frequency in some human geographic populations (*Kidd et al., 2007*); thus, it would be interesting to determine whether the loss of A3B correlates with increased frequency of de novo L1 insertions.

APOBEC3 activity has been implicated in genome hyper-mutation in numerous cancers (*Roberts et al., 2013*; *Burns et al., 2013a*, *2013b*). It is interesting to speculate that APOBEC3 expression in cancers may restrict potentially deleterious L1 retrotransposition events, yet may exact a mutagenic cost by contributing to genome hyper-mutation (*Narvaiza et al., 2012*). Indeed, our results suggest that in addition to TPRT, A3A may deaminate any transiently exposed single-strand region of genomic DNA (e.g., those arising at replication forks or during transcription). This hypothesis is consistent with recent reports indicating that resected single-strand DNA, around DNA breaks, is susceptible to hyper-mutation by APOBEC deamination (*Nik-Zainal et al., 2012*; *Roberts et al., 2012*; *Taylor et al., 2013a*).

## Materials and methods

### Plasmids

All plasmids were grown in DH5α (F– Φ80*lacZ*ΔM15 Δ(*lacZYA-arg*F) U169 *rec*A1 *end*A1 *hsd*R17 (rK–, mK+) *pho*A *sup*E44 λ–*thi*-1 *gyr*A96 *rel*A1) competent *E. coli* (Invitrogen, Carlsbad, CA). Competent cells were prepared in house as described in *Inoue et al. (1990)*. Plasmid preparations were generated using the Qiagen Plasmid Midi Kit (Qiagen, Germany) according to the manufacturer's protocol.

### APOBEC3 expression constructs

The pK_βarr, pK_A3A, pK_A3B, pK_A3B_Nterm, and pK_A3B_CS expression plasmids were described previously (*Bogerd et al., 2006a*). pK_A3B_Nterm expresses only the N-terminal half of the A3B protein and pK_A3B_CS contains full length A3B with an inactivating point mutation in the C-terminal deaminase active site; both constructs lack deaminase activity in a bacterial mutator assay (*Bogerd et al., 2006b*). The previously described deaminase-deficient A3A mutant (A3A-C106S [*Chen et al., 2006*]) was subcloned into the **pK expression vector** using *Hind*III and *Xho*I restriction sites.

### LINE expression constructs

#### pJM101/L1.3

pJM101/L1.3 was described previously (*Dombroski et al., 1993*; *Sassaman et al., 1997*) and consists of the pCEP4 backbone (Life Technologies, Carlsbad, CA) containing a full-length copy of the L1.3 element (accession # L19088) and the *mneoI* indicator cassette.

#### pDK101

pDK101 was described previously (*Kulpa and Moran, 2005*) and consists of pJM101/L1.3 modified by PCR mutagenesis to contain the T7 *gene 10* epitope tag at the C-terminus of ORF1p.

#### pDK135

pDK135 was described previously (*Kulpa and Moran, 2005*) and is identical to pDK101, except it contains the $D_{702}A$ mutation in the putative ORF2p reverse transcriptase active site (*Wei et al., 2001*).

#### pJJH230A/L1.3

pJJH230A/L1.3 was described previously (*Kopera et al., 2011*) and is similar to pJM101/L1.3, except it contains the *mblastI* indicator cassette (*Morrish et al., 2002*) in the 3′ UTR instead of *mneoI* and the $H_{230}A$ mutation in the ORF2p endonuclease domain (*Wei et al., 2001*).

#### pKS101/L1.3/sv+

pKS101/L1.3/sv+ consists of the pBSKS-II backbone (Agilent Technologies, Santa Clara, CA) containing a full-length L1.3 element with the *mneoI* indicator cassette in the 3′UTR followed by the SV40 late polyadenylation signal.

### pJM102/L1.3

pJM102/L1.3 was described previously (*Morrish et al., 2002*) and is identical to pJM101/L1.3, except it lacks the native L1.3 5'UTR.

### pCEP4/L1SM

pCEP4/L1SM was described previously (*Han and Boeke, 2004*) and consists of a synthetic mouse L1 sharing the same amino acid sequence as L1$_{spa}$ (*Naas et al., 1998*), but with 24% of its nucleotide sequence replaced for optimal GC-richness. It contains the *mneoI* indicator cassette in the 3'UTR and is cloned into the pCEP4 backbone.

### pCEP4/TG$_F$21

pCEP4/TG$_F$21 was described previously (*Goodier et al., 2001*) and consists of a natural mouse element with the *mneoI* indicator cassette in the 3'UTR cloned into the pCEP4 backbone.

### pCEP4/ZfL2-2

pCEP4/ZfL2-2 was described previously (*Sugano et al., 2006*) and consists of a zebrafish LINE-2 element cloned into the pCEP4 backbone. The ZfL2-2 3'UTR is cloned 3' of the *mneoI* cassette.

### pAD2TE1

pAD2TE1 was described previously (*Doucet et al., 2010*) and is similar to pDK101, except it contains a TAP epitope tag on the C-terminus of ORF2p as well as the T7 *gene 10* epitope tag on the C-terminus of ORF1p.

### pAD136

pAD136 was described previously (*Doucet et al., 2010*) and is identical to pAD2TE1, except it contains the H$_{230}$A mutation in the ORF2p endonuclease domain.

### pJM140/L1.3/$\Delta^2$/k7

pJM140/L1.3/$\Delta^2$/k7 was described previously (*Gilbert et al., 2005*) and contains the L1.3 element cloned into the pCEP4 backbone, but lacks the CMV promoter and the SV40 poly A signal. The *mneoI* indicator cassette and a ColE1 bacterial origin of replication are inserted in the L1.3 3'UTR.

### pLGCX vector

pLGCX vector was described previously (*Kaiser and Emerman, 2006*) and is a variant of the LNCX retroviral vector (*Miller and Rosman, 1989*) in which the neomycin phosphotransferase gene has been replaced by GFP.

### pLGCX/UGI

pLGCX/UGI was described previously (*Kaiser and Emerman, 2006*) and consists of the LGCX vector containing a uracil glycosylase inhibitor (UGI) gene from *Bacillus subtilis* bacteriophage PBS2 codon-optimized for expression in human cells (hUGI).

### pU6i NEO

pU6i NEO is a pBSKS-based plasmid with the neomycin phosphotransferase (*NEO*) gene from pEGFP-N1 (Clontech, Mountain View, CA) introduced into the backbone. The multi-cloning site contains the U6 promoter. To generate linearized plasmid for control transfections, pU6i *NEO* was digested with *Bgl*II (New England Biolabs, Ipswich, MA), which does not disrupt *NEO* gene expression, and run on an agarose gel to confirm linearization. The restriction digest reactions were purified using the QIAquick Gel Extraction Kit (Qiagen).

## Cell Culture

All human cultured cell lines employed in this study were authenticated by STR comparative analysis, performed by Genetica DNA Laboratories Inc., Burlington, NC. The Chinese Hamster Ovary cell lines were used and verified in previous studies (*Morrish et al., 2002*, *2007*).

### HeLa cells

HeLa cells were reported in the following refs. (*Moran et al., 1996*; *Wei et al., 2000*). They were grown at 37°C in Dulbecco's modified Eagle medium (DMEM) (Invitrogen) supplemented with 10% fetal bovine calf serum (FBS) (Invitrogen) and 1X penicillin/streptomycin/glutamine (Invitrogen) in a humidified incubator in 7% $CO_2$.

## Chinese Hamster Ovary cells

Chinese Hamster Ovary cells were reported in the following refs. (*Morrish et al., 2007*, *2002*). They were grown at 37°C in DMEM-low glucose medium (Invitrogen) supplemented with 10% FBS (Invitrogen), 1X penicillin/streptomycin/glutamine (Invitrogen), and 1X non-essential amino acids (Invitrogen) in a humidified incubator in 7% $CO_2$.

## U2OS_UGI and U2OS control cells

U2OS_UGI and U2OS control cells were reported in ref. (*Landry et al., 2011*). They were grown at 37°C in DMEM (Invitrogen) supplemented with 10% FBS (Invitrogen) and 1X penicillin/streptomycin/glutamine (Invitrogen) in a humidified incubator in 7% $CO_2$.

## L1 retrotransposition assays

HeLa and U2OS cell retrotransposition assays were carried out as previously described (*Moran et al., 1996*; *Wei et al., 2000*). Cells were plated in 6-well dishes (BD Biosciences, San Jose, CA), T-75 flasks (BD Biosciences), or 10 cm dishes (BD Biosciences or Corning, Corning, NY). For retrotransposition assays employing human L1.3-based constructs (including pJM101/L1.3, pDK101, pKS101/L1.3/sv+, and pJM102/L1.3), approximately $1 \times 10^5$ cells were plated per T-75 flask or 10 cm dish, and approximately $1 \times 10^3$ cells were plated per well in a six-well dish. For assays employing pCEP4/L1SM, approximately $2 \times 10^4$ cells were plated per T-75 flask. For assays employing $TG_F21$ and Zfl2-2, approximately $4 \times 10^5$ cells were plated per T-75 flask. Cell plating densities were optimized so that positive controls would yield a quantifiable number of G418-resistant foci (~100–500 foci in a six-well dish and ~500–2500 foci per 10 cm dish or T-75 flask). The reduction of G418-resistant foci in the presence of A3A was determined to be reliable and reproducible. Approximately eighteen hours after plating, transfections were carried out using the FuGene 6 transfection reagent (Roche, Switzerland) and Opti-MEM (Life Technologies), according to the protocol provided by the manufacturer (3 µl FuGene and 97 µl Opti-MEM per one µg of DNA transfected). Media were replaced the following day. Approximately 72 hr post-transfection, cells were subjected to selection with 400 µg/ml G418 (Life Technologies). G418 selection was carried out for 12–14 days and the selection media were replaced every other day. Colonies were washed with 1X phosphate buffered saline (PBS) (Gibco, Carlsbad, CA), fixed with 2% paraformaldehyde/0.4% glutaraldehyde, and stained with 0.1% crystal violet solution.

CHO cell retrotransposition assays were carried out in 4364a and XR-1 cell lines as previously described (*Morrish et al., 2002*). Approximately $1 \times 10^5$ cells were plated per T-75 flask or $2 \times 10^4$ cells per well of a six-well dish. 8 hr later, transfections with wild-type (pAD2TE1) and endonuclease mutant (pAD136) retrotransposition indicator plasmids were carried out using the FuGene 6 transfection reagent and Opti-MEM according to manufacturer's protocol (3 µl FuGene and 97 µl Opti-MEM per one µg of DNA transfected). Media were replaced the following day. Approximately 72 hr post-transfection, cells were subjected to selection with 400 µg/ml G418. Selection was carried out for 12–14 days and the selection media were replaced every other day. Colonies were washed with 1X PBS, fixed with 2% paraformaldehyde/0.4% glutaraldehyde, and stained with 0.1% crystal violet solution.

For toxicity control transfections, which were carried out in parallel with each retrotransposition assay, HeLa and CHO cells were plated and transfected as described above, substituting pU6i NEO plasmid for the L1 retrotransposition indicator cassette. As in the retrotransposition assays, selection with 400 µg/ml G418 was initiated approximately 72 hr post-transfection and carried out for 12–14 days. The selection media were replaced every other day. Colonies were fixed and stained as described for the retrotransposition assays. The percent of G418 colonies obtained from A3A/pU6i NEO co-transfections relative to control (β-arrestin/pU6i NEO) co-transfections was calculated and used to normalize the number of G418-resistant colonies obtained in the corresponding A3A/L1 and control β-arrestin/L1 retrotransposition assays. In this way, any nonspecific toxicity caused by A3A expression is accounted for, and the percent inhibition reported for each assay reflects specific inhibition of L1 retrotransposition by A3A.

## Recombinant A3A protein expression and purification

WT A3A and deaminase-deficient A3A-C106S cDNAs were inserted into pET28 histidine-tag expression plasmids and expressed in *E. coli*, strain C43 (F– ompT hsdSB (rB- mB-) gal dcm (DE3)) (Lucigen, Middleton, WI). Recombinant protein was produced by expression at 37°C in LB plus

kanamycin for 3 to 5 hr after induction with 1 mM IPTG. Bacterial cultures were pelleted and cell pellets were frozen prior to lysis. Cell pellets were resuspended in lysis buffer (50 mM Tris–HCl [pH8], 200 mM KCl, 10% glycerol, 1 mM DTT, 35 mM imidazole, and Complete protease inhibitor [Roche]) and lysed using a fluidizer (Microfluidics, Westwood, MA). The soluble (supernatant) fraction was separated by centrifugation at 40,000×$g$ for 1 hr at 6°C. Recombinant protein from the supernatant was purified by binding to Ni-Sepharose Fast Flow (GE Healthcare, United Kingdom), washed with lysis buffer, and eluted with lysis buffer containing 0.5M imidazole. Protein was concentrated using Amicon Ultracell-10K filters (Millipore, Billerica, MA), further purified by gel filtration on a HiLoad 16/60 Superdex 200 prep grade column (Amersham, United Kingdom), and dialyzed in Slide-A-Lyzer 10K MWCO cassettes (Pierce, Rockford, IL) in dialysis buffer (50 mM Tris–HCl [pH 8], 200 mM KCl, 10% glycerol, and 1 mM DTT). The proteins then were concentrated using Amicon Ultracell-10K filters (Millipore). Protein purification was monitored by SDS-PAGE, Coomassie blue staining, and immunoblotting with antibodies against His-tag (Sigma, St. Louis, MO) and APOBEC3A (*Narvaiza et al., 2009*). UV-spectrophotometry and ImageJ software were used to quantify protein levels.

## In vitro A3A deaminase assays

To determine rA3A deaminase activity in UDG-dependent deaminase assays, increasing concentrations of rA3A or rA3A_C106S were incubated with a 5′-end fluorescein isothiocyanate (FITC) labeled single-strand deoxyoligonucleotide (0.4 μM) in a final reaction volume of 30 μl containing: 40 mM Tris–HCl (pH 8.0) 10% glycerol, 40 mM KCl, 50 mM NaCl, 5 mM EDTA, and 1 mM DTT. The reactions were incubated at 37°C for 4–8 hr, stopped by heating to 90°C for 5 min, cooled on ice, and then centrifuged at 10,000×$g$ for 1 min. 20 μl of the supernatant were then incubated with uracil DNA glycosylase (UDG, New England Biolabs) in buffer containing 20 mM Tris–HCl (pH 8.0) and 1 mM DTT for 1 hr at 37°C. The resultant products were treated with 150 mM NaOH for 30 min at 37°C, followed by incubation at 95°C for 5 min, and chilled at 4°C for 2 min. The samples were separated using 15% TBE/urea-PAGE. Gels were directly analyzed using a FLA-5100 scanner (Fuji, Japan). The PAGE-purified single-strand DNA oligonucleotide (Invitrogen) used for the deaminase assays (FITC-TCA) contains a single cytosine in the A3A specific target trinucleotide, 5′-TCA:

   (FITC-5′-TATTATTATTATTATTATT**C**ATTTATTTATTTATTTATTT-3′)

For deaminase assays on double-strand DNA substrates, the target FITC-TCA oligonucleotide was pre-incubated with either a complementary (asOligo) or non-complementary (ns) oligonucleotide at the ratios indicated in *Figure 2—figure supplement 1C*.

## LEAP assay

Preparation of L1 RNPs and LEAP assays were carried out generally as previously described in *Kulpa and Moran (2006)*.

### Oligonucleotide primer sequences

Integrated DNA Technologies (IDT, Coralville, IA) synthesized the oligonucleotides used in the studies below.

   LEAP adapter 5np1, purified by high-performance liquid chromatography (HPLC):

   5′-GATGGATGATGAATAAAGTGTGGGATGATCATGATGTATGGATAGGTTTTTTTTTTTTT-3′
   5np1 outer: 5′-GATGGATGATGAATAAAGTG-3′
   L1 3′ end: 5′-GGGTTCGAAATCGATAAGCTTGGATCCAGAC-3′ (*Kulpa and Moran, 2006*)

### Preparation of L1 RNPs

Approximately $6 \times 10^6$ HeLa cells were plated per T-175 flask (BD Biosciences). Approximately eighteen hours later, they were transfected with 30 μg of L1 plasmid DNA using FuGene 6 transfection reagent and Opti-MEM according to manufacturer's protocol (3 μl FuGene and 97 μl Opti-MEM per μg of DNA transfected). Approximately 72 hr post-transfection, cells were subjected to selection with 200 μg/ml Hygromycin B (Life Technologies). Hygromycin media were replaced daily for 9–11 days. Untransfected control HeLa cells were plated 2–3 days before harvest day. On harvest day, cells were washed three times with 10 ml cold 1 × PBS and collected by scraping into 10 ml cold 1 × PBS. Cells were pelleted in a 15 ml conical tube (BD Biosciences) at 3000 × $g$ for 5 min at 4°C. The PBS was removed and the cells were lysed for 15 min on ice in 1 ml lysis buffer (1.5 mM KCl, 2.5 mM MgCl$_2$, 5 mM Tris–HCl [pH 7.4], 1% wt/vol deoxycholic acid, 1% wt/vol Triton X-100,

and 1x Complete EDTA-free protease inhibitor cocktail [Roche]). Following lysis, cell debris was pelleted by centrifugation at 3000×*g* for 5 min at 4°C and the whole-cell lysate was transferred to a new tube.

## Isolation of RNPs by ultracentrifugation

Whole-cell lysate (1 ml) was centrifuged through a sucrose cushion consisting of 8.5% and 17% wt/vol sucrose in 80 mM NaCl, 5 mM MgCl$_2$, 20 mM Tris–HCl (pH 7.5), 1 mM DTT, and 1 × Complete EDTA-free protease inhibitor cocktail (Roche). Cells were ultracentrifuged for 2 hr at 178,000×*g* at 4°C. The resulting pellet was resuspended in 50–100 µl (depending on pellet volume) in 1 × Complete EDTA-free protease inhibitor cocktail (Roche). Protein concentration was determined by Bradford assay (Bio-Rad, Hercules, CA) and the sample was brought to a final concentration of 1.5 mg/ml. The portion of the sample intended for LEAP reactions was diluted to 50% vol/vol glycerol, aliquoted, flash-frozen on dry ice/ethanol, and stored at −80°C. Additional portions were reserved for RNA extraction.

## RNA isolation and RT-PCR

RNA was extracted from RNP preparations using the RNeasy Mini Kit (Qiagen) following the manufacturer's instructions, omitting the cell lysis step and including on-column DNase treatment. RNA was quantified using a Nano-Drop spectrophotometer and diluted to 0.5 µg/µl. RT-PCR was carried out on 0.5 µl of purified RNA, using MMLV reverse transcriptase (Promega, Madison, WI) and 0.8 µM LEAP adapter (5np1) at 42°C for 30 min. Reactions containing 100–300 ng of rA3A or rA3A_C106S protein in 10% glycerol were included in the MMLV RT reaction. For control reactions containing 'heat-killed' rA3A, rA3A samples were incubated at 100°C for 15 min prior to inclusion in the MMLV RT reaction. Following the MMLV RT reaction, reaction mixtures were incubated at 100°C for 15 min to denature rA3A protein. An aliquot (0.5 µl) of the MMLV-RT reaction product was PCR amplified with *PfuTurbo C$_x$* Hotstart DNA polymerase (Agilent) using the following cycling conditions: 95°C for 2 min, followed by 35 cycles of 30 s at 95°C, 30 s at 58°C, and 30 s at 72°C, with a final extension time of 7 min at 72°C. PCR products were visualized on 2% agarose gels.

## LEAP reactions

One µl of 0.75 µg/µl (50% vol/vol glycerol) RNP sample was incubated with 50 mM Tris–HCl (pH 7.5), 50 mM KCl, 5 mM MgCl$_2$, 10 mM DTT, 0.4 µM LEAP primer, 20 units of RNasin (Promega), 0.2 mM dNTPs (Invitrogen), and 0.05% vol/vol Tween 20 in a final volume of 50 µl for 30 min at 37°C. A 'no RNP' reaction was prepared as a negative control. For A3A containing reactions, 5–100 ng of WT rA3A or deaminase-deficient rA3A_C106S protein in 10% glycerol were included in the LEAP reaction. For RNase H-containing reactions, 2 units of recombinant RNase H (Invitrogen) were included in the LEAP reaction. For control reactions containing 'heat-killed' rA3A, rA3A samples were incubated at 100°C for 15 min prior to inclusion in the LEAP reaction. Following the L1 RT reaction, reactions were incubated at 100°C for 15 min to denature rA3A protein. An aliquot (1 µl) of LEAP product was then included in the PCR amplification step using *PfuTurbo C$_x$* Hotstart DNA polymerase (Agilent) under the following conditions: 95°C for 2 min, followed by 35 cycles of 30 s at 95°C, 30 s at 58°C, and 30 s at 72°C, with a final extension time of 7 min at 72°C. PCR products were visualized on 2% agarose gels.

## Product characterization

PCR products were excised from agarose gels and purified using the QIAquick Gel Extraction Kit (Qiagen). Products were cloned into ZERO Blunt PCR cloning vector (Invitrogen), transformed, and plasmid DNA recovered by mini-prep using the Wizard Plus SV Minipreps DNA Purification System (Promega). Individual clones were then sequenced. Sequence alignments were produced using MegAlign software from the Lasergene DNASTAR suite.

## Characterization of L1 retrotransposition events

### Oligonucleotide primer sequences

Integrated DNA Technologies (IDT) synthesized the oligonucleotides used in the studies below:

NEOasReco: 5′-TCTATCGCCTTCTTGACGAG-3′
Rescue3seq: 5′-ACTCACGTTAAGGGATTTTGGTCA-3′
PolyAseq: 5′-AAAAAAAAAAAAAAAAAAAAAABN-3′

## Generation of clonal Cell lines

Approximately $2 \times 10^5$ HeLa cells were plated per 15 cm dish (BD Biosciences) and transfected using FuGene 6 and Opti-MEM with 3 µg each of pLGCX/UGI, an A3A expression vector, and pJM140/L1.3/$\Delta^2$/k7, according to manufacturer's protocol (3 µl FuGene and 97 µl Opti-MEM per one µg of DNA transfected). For U2OS_UGI cells, approximately $5 \times 10^5$ cells were plated per 15 cm dish and transfected with either 2 µg of an A3A, A3A_C106S, or β-arrestin expression plasmid and 8 µg of pJM140/L1.3/$\Delta^2$/k7, using FuGene 6 and Opti-MEM according to manufacturer's protocol (3 µl FuGene and 97 µl Opti-MEM per one µg of DNA transfected). Selection with 400 µg/ml of G418 was initiated approximately 72 hr post-transfection and carried out for 14 days. G418 media were replaced every other day. On day 14, individual colonies were manually picked into individual wells of 12-well culture dishes (BD Biosciences). These colonies were expanded for 2–3 weeks to establish clonal cell lines. Cells from confluent T-75 flasks (HeLa) or 15 cm dishes (U2OS) of each cell line were harvested by trypsinization and genomic DNA was prepared using the Qiagen Blood and Cell Culture DNA Midi Kit, according to manufacturer's instructions.

## Recovery of L1 insertions

Insertions were recovered generally as previously described (*Gilbert et al., 2002*, *2005*). Briefly, 8 µg of genomic DNA were digested with a single restriction enzyme (*Hind*III, *Ssp*I, *Bgl*II, or *Bam*HI: New England Biolabs; *Bcl*I: Promega) overnight at 37°C. The following morning, an additional 2 µl of the respective restriction enzyme were added to the digest, and the reactions were incubated at 37°C for 2 hr. Digestion reactions were heat-inactivated (65°C, 15 min). In the case of *Bgl*II digestions, the Wizard DNA Clean-Up Kit (Promega) was used to inactivate the enzyme. The entire digest was then ligated with 3200 units of T4 DNA ligase (New England Biolabs) under dilute conditions (500 µl total volume) overnight at 16°C. The next morning, an additional 2 µl of T4 DNA ligase were added to the reaction and incubated at room temperature for 4 hr. Ligations were concentrated on an Amicon Ultra-0.5 Centrifugal Filter Unit with Ultracel-100 membrane (Millipore) at 8000×$g$ for 5 min. The membrane then was washed by applying 500 µl dH$_2$O, and spinning for 5 min at 8000 × $g$. An aliquot (10–100 µl) of concentrated DNA was recovered with a 10 s spin at 8000 × $g$. The entire ligation then was added to 500 µl of XL-10 gold ultra-competent *E. coli* (Stratagene, La Jolla, CA: prepared in-house as described in *Inoue et al. (1990)*) and incubated on ice for 1–3 hr. Transformations were heat-shocked at 42°C for 38–45 s and allowed to recover on ice for 2 min. 2 ml of warm (37°C) LB (no antibiotic) were added to each transformation. Transformations were incubated overnight at room temperature on an orbital shaker at 100 rpm. The next morning, transformations were pelleted (300 × $g$) and gently resuspended in 500 µl fresh LB. About 400 µl of this resuspension were plated on a 15 cm Kanamycin plate (30 µg/ml) and the remaining ~100 µl were used to seed a 2 ml liquid culture (30 µg/ml Kanamycin). Plates and liquid cultures were incubated for 18–24 hr at 37°C. DNAs were prepared from 2 ml cultures using the Wizard Plus SV Minipreps DNA Purification System (Promega). Individual colonies were picked from 15 cm plates to inoculate 2 ml cultures (30 µg/ml Kanamycin), grown overnight at 37°C, and plasmid DNA was prepared by mini-prep using the Wizard Plus SV Minipreps DNA Purification System (Promega) the following day. Plasmid DNA recovered from mini-preps was digested with the original enzyme used for recovery to confirm intramolecular ligation. To characterize flanking genomic DNA, recovered insertions were sequenced using primers annealing to the 5′ (NEOasReco) and 3′ (Rescue3seq) ends of the NEO_ColE1 recovery cassette, as well as an oligo dA primer (polyAseq). The genomic locations of insertions were determined by aligning flanking sequence to the human genome (February 2009; GRCh37/hg19), using the BLAT function of the UCSC genome browser (http://genome.ucsc.edu/cgi-bin/hgBlat; *Kent, 2002*).

## Acknowledgements

We thank Dr José Luis Garcia-Perez and members of the Moran lab for critical readings of this manuscript. We thank Ms. Nancy Leff for editorial assistance. We thank Dr H Kazazian for providing the pCEP4/TG$_F$21 construct, Dr J Boeke for providing the pCEP4/L1SM construct, and Drs M Kajikawa and N Okada for providing the pCEP4/ZfL2-2 construct. We thank Dr M Emerman for providing the pLGCX vector and pLGCX_UGI plasmids. We also thank Dr C Tzitzilonis for technical assistance and helpful discussion on rA3A purification. We thank Mr S Morell for technical assistance with the L1 insertion recovery assay.

## Additional information

### Competing interests

JVM: is an inventor on patent 6,150,160. Kazazian, H.H., Boeke, J.D., Moran, J.V., and Dombroski, B.A.: Compositions and methods of use of mammalian retrotransposons. The other authors declare that no competing interests exist.

### Funding

| Funder | Grant reference number | Author |
|---|---|---|
| Howard Hughes Medical Institute | | John V Moran |
| National Institutes of Health | GM060518 | John V Moran |
| National Institutes of Health | T32-GM07544 | Sandra R Richardson |
| National Institutes of Health | AI074967 | Matthew D Weitzman |
| Instituto de Salud Carlos III/Consejo Superior de Investigaciones Cientificas | | Iñigo Narvaiza |
| Lynn Streim Postdoctoral Endowment Fellowship | | Iñigo Narvaiza |

The funders had no role in study design, data collection and interpretation, or the decision to submit the work for publication.

### Author contributions

SRR, IN, MDW, Conception and design, Acquisition of data, Analysis and interpretation of data, Drafting or revising the article; RAP, Acquisition of data, Analysis and interpretation of data; JVM, Conception and design, Analysis and interpretation of data, Drafting or revising the article

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
