## [Decision Letter]

Thank you for sending your work entitled “APOBEC3A Deaminates Transiently Exposed Single-Strand DNA During LINE-1 Retrotransposition” for consideration at eLife. Your article has been very favorably evaluated by a Senior editor, a Reviewing editor, and 3 reviewers.

The following individuals responsible for the peer review of your submission have agreed to reveal their identity: Anne Ferguson-Smith (Reviewing editor), and Harmit Malik and Nina Papavasiliou (peer reviewers). A third reviewer remained anonymous.

The reviewers noted: 'Well (and cleverly) done – no experimental concerns'. 'I find the data to be compelling and very clearly presented. This is a short, crisp and high impact study'. 'The manuscript is very comprehensive and well written. Materials and methods are well described, and the figures very clear. I was a real pleasure to review this nice paper'.

We all agree that the manuscript is suitable for publication and we are delighted to accept it for *eLife*. We have made some minor comments for your consideration, which we feel would clarify some points and improve the structure and readability of the manuscript.

Minor comments:

Although the choice of format is at the discretion of the authors, perhaps it is not in the best interest of the paper to have a combined Results and Discussion section. Indeed, although several small discussion points motivate the next experiment, the true Discussion begins with the paragraph “We propose a straightforward model....” and the model Figure 4. So, please consider splitting the Results and Discussion.

In addition, the Discussion might include three further points:

(a) The few L1 retrotransposition events that occur in the presence of A3A that are devoid of obvious deamination - is the model that these reflect cases where the TPRT events were beyond the 'saturation limit' or restriction capacity of A3A?

(b) Are there differences in A3A and A3B (which has recently become quite prominent in cancer mutagenesis) which have similar preferences (and anti anti-L1 activity) but perhaps slightly different cellular localization? Finally, it may bear mentioning that whereas A3B is expressed in germline cells, A3A is not (I think).

(c) Is this something that actually happens in real life? For instance, if A3A (and other A3's for that matter) have anti-retrotransposon activities in unmanipulated systems, then it follows that their absence would correlate with increased transposition. Mouse has a single A3 and mA3 deficiency, though possibly anti-retroviral, has not led to appreciable defects in transposable element restriction (though detecting such events is not always easy, a fact which this reviewer certainly appreciates). Perhaps the authors can comment on this.

---

## [Author Response]

*Although the choice of format is at the discretion of the authors, perhaps it is not in the best interest of the paper to have a combined Results and Discussion section. Indeed, although several small discussion points motivate the next experiment, the true Discussion begins with the paragraph “We propose a straightforward model....” and the model*
Figure 4*. So, please consider splitting the Results and Discussion*.

We agree with the reviewers’ suggestion. As requested, we have now separated the Results and Discussion sections. We agree that this change was in the best interest of the paper.

*In addition, the Discussion might*
*include three further points:*

*(a) The few L1 retrotransposition events that occur in the presence of A3A that are devoid of obvious deamination - is the model that these reflect cases where the TPRT events were beyond the 'saturation limit' or*
*restriction capacity of A3A?*

We thank the reviewers for their scholarly comments. As requested we have now included a new paragraph that discusses possibilities for why some L1 retrotransposition events are devoid of deamination. Specifically, we state the following in the Discussion section of the manuscript:

“The over-expression of A3A does not abolish L1 retrotransposition completely, but reduces it to ∼30% of control levels. Moreover, some L1 retrotransposition events generated in the presence of A3A and UGI lacked evidence of deamination. Why and how these L1 retrotransposition events evade A3A-mediated deamination requires further study. However, it is possible that ectopic A3A expression is too low to efficiently target L1 TPRT intermediates in some cells or that L1 second-strand cDNA synthesis occasionally is completed before A3A can gain access to TPRT intermediates to deaminate single-strand L1 cDNAs.”

*(b) Are there differences in A3A and A3B (which has recently become quite prominent in cancer mutagenesis) which have similar preferences (and anti anti-L1 activity) but perhaps slightly different cellular localization? Finally, it may bear mentioning that whereas A3B is expressed in germline cells, A3A is not (I think)*.

Again, we thank the reviewers for their comments. As requested, we have highlighted differences between the expression patterns of A3A and A3B in the Discussion section of the manuscript. We also have highlighted that both A3A and A3B can localize to the nucleus. Specifically, we have added text to our Discussion (paragraph starting” Mechanistic insights gained from studying deaminase-dependent A3A-mediated L1 inhibition could be applicable to retroelement inhibition by other A3 proteins”).

We further agree with the statement that A3B has become quite prominent in cancer mutagenesis and we have added the text starting “APOBEC3 activity has been implicated in genome hyper-mutation in numerous cancers (59; 6; 7).”

*(c) Is this something that actually happens in real life? For instance, if A3A (and other A3's for that matter) have anti-retrotransposon activities in unmanipulated systems, then it follows that their absence would correlate with increased transposition. Mouse has a single A3 and mA3 deficiency, though possibly anti-retroviral, has not led to appreciable defects in transposable element restriction (though detecting such events is not always easy, a fact which this reviewer certainly appreciates). Perhaps the authors can comment on this*.

Again, we thank the reviewers for their comments. Because A3A expression is largely restricted to peripheral blood lymphocytes, we agree that it may not restrict germ line L1 retrotransposition events in “real life.” However, we posit that mechanistic insights gained from studying deaminase-dependent A3A-mediated L1 inhibition could be applicable to retroelement inhibition by other A3 proteins. For example, and as detailed in our response above, we highlight that the related A3B cytidine deaminase protein (which localizes to the nucleus, is expressed in human embryonic stem cells (hESCs), and can inhibit the retrotransposition of engineered L1s in hESCs) may restrict germline L1 retrotransposition events. Moreover, we have provided a brief discussion about how differences in A3B expression levels may, in part, contribute to differences in L1 activity reported in human and non-human primate induced pluripotent stem cells. Finally, we raise the possibility that differences in the presence/absence of A3B in humans may contribute to differences in L1 retrotransposition activity.

To the best of our knowledge, no one has conducted an in depth study of L1 retrotransposition activity in A3 deficient mice; thus, we did not discuss this point in our study.